# Peer review of "Quadruplex Droplet Digital PCR Assay for Screening and Quantification of SARS-CoV-2"

_ijms, 2024, doi:10.3390/ijms25158157_

Round 1

Reviewer 1 Report

Comments and Suggestions for Authors

The present study establishes a quadruplex droplet digital PCR nucleic acid detection system for 2019-nCoV. The system's limit of detection (LOD), specificity, and linearity are evaluated and compared with those of the single-fold digital PCR system. I have four suggestions to enhance its professionalism.

1.The detection of qRT-PCR and ddPCR was not compared in this paper. Please provide data if relevant comparison has been made.

2.Why selected E, N, RdRp and ORF1a/b genes as target genes?

3.Please describe the method for determining the results of quadruplex droplet digital PCR.

4.Whether to evaluate the linearity of the detection results of high-concentration samples (> 20,000 copies/reactions) of the multiplex digital PCR system? If so, please provide relevant data.

Comments on the Quality of English Language

The article's structure is logical and clear. 

Reviewer 2 Report

Comments and Suggestions for Authors

I really don't have much to say about this article. The work presented offers a new approach to the diagnosis of Covid-19, which until now has always been a public health problem, and the technique is very interesting.

Just two or three small things :

Line 70. Can the author have more information on the swab samples received? For example, what were their selection criteria?

Is 4-plex (line 103, 124, 351-353, Table1 and 4) used for simplex or quadruplex? This has not been clarified beforehand and is causing confusion. 

From line 320 to 324, the Figure legend section is empty. Was this an oversight or an error? To be checked

Table 4 appears to be cut off towards the end, but this also needs to be checked.
